# Simple mapping-based quantification of a mock microbial community using total RNA-seq data

**Shigeharu Moriya** [ID] *

Environmental Metabolic Analysis Research Team, Center for Sustainable Resource Science, RIKEN Institute, Yokohama, Kanagawa, Japan

* smoriya@riken.jp

## Abstract

Most microbes in the natural environment are difficult to cultivate. Thus, culture-independent analysis for microbial community structure is important for the understanding of its ecological functions. An immense ribosomal RNA sequence collection is available from phylogenetic research on organisms in all domains. These sequences are available for use in genetic research. However, the amplicon-seq process using PCR requires the construction of a sequence library. Construction can introduce bias into quantitative analyses, and each domain of species needs its own primer set. Total RNA sequencing has the advantage of analyzing an entire microbial community, including bacteria, archea, and eukaryote, at once. Such analysis yields large amounts of ribosomal RNA sequences that can be used for analysis without PCR bias. Evaluation using total RNA-seq for quantitative analysis of microbial communities and comparison with amplicon-seq is still rare. In the present study, we developed a mapping-based total RNA-seq analysis to obtain quantitative information on microbial community structure and compared our results with ordinary amplicon-seq methods. We read total RNA sequences from a commercially available mock community (ATCC MSA-2003) and divided reads into small subunit ribosomal RNA (ssrRNA) origin reads and others, such as mRNA origin reads. We then mapped ssrRNA origin reads on annotated assembled contigs and obtained quantitative results under several analysis strategies. Removal of low complexity sequences, sorting ssrRNA with paired-in mode, and performing homology-based taxonomical assignments (BLAST+ or vsearch) showed superior outcomes to other strategies. Results with this approach showed a median relative abundance among ten mock community members of ~10%; ordinary amplicon-seq showed a much lower percentage. Thus, total RNA-seq can be a powerful tool for analyzing microbial community structure and is not limited to analyzing gene expression profiling of microbiomes.

## Introduction

Understanding ecological services of microbial communities require knowledge of community composition. Most microbes are difficult to cultivate, yet microbial community structure

**Data Availability Statement:** All sequence data are deposited in DDBJ DRA, accession number DRA009985. (https://ddbj.nig.ac.jp/DRASearch/submission?acc=DRA009985).

**Funding:** This work was supported by the Ministry of Education, Culture, Sports, Science and Technology Grant-in-Aid for Scientific Research on Innovative Areas, JP3308, and the RIKEN integrated symbiology program.

**Competing interests:** Author have no competitive interests.

analysis is an important tool for investigation of environmental microbial activity. As an alternative to cultivation, a molecular phylogenetic approach is widely used. RNA and DNA can be extracted from environmental samples without cultivation, and PCR with specific "bar-code" gene(s) can be used for phylogenetic classification of microbes. Accumulation of molecular phylogenetic information allows molecular classification based on "bar-code" gene sequence comparisons with molecular phylogeny.

Small subunit ribosomal RNA (ssrRNA) is a well-established "bar-code" gene for taxonomic identification because it is conserved among organisms with the same biological function. Early molecular phylogenetic work with ssrRNA sequences uncovered three life domains [1] and an unexpected variety of not-yet-cultivated microorganisms [2]. Long-term accumulation of ssrRNA sequences in the context of phylogenetic and taxonomic investigation led to the development of large public ribosomal RNA (rRNA) databases such as RDP, Greengenes, and Silva [3–5]. Using this common "bar-code" among all domain organisms, we can identify microbes by similarity with sequences stored in these databases, and we can classify new species using of phylogenetic analyses with those sequences.

Modern molecular biology and high throughput sequencing provide the opportunity to comprehensively evaluate microbial communities. Microbial rRNA sequences can be amplified from any environmental or clinical samples and can be read by a massively parallel sequencer [6–10]. rRNA databases can then be used to define microbial community structure by comparison among "bar-code" genes. This technique is called "amplicon-seq" and is widely used in microbial ecology [8–10].

Amplicon-seq is a powerful tool but has several weak points. rRNA has several conserved sequences, e.g., the stem region, yet universal primers across different domains are difficult to create. Hence, amplicon-seq should be applied separately among domains—bacteria, archaea, or eukaryotes. Furthermore, PCR introduces bias because of sequence differences among microbes. In some cases, contamination with environmental DNA is a problem. For example, frozen soil sample may include not only live microbes but also microbes destroyed by the freezing process. DNA from dead microbes can cause noise, for example, when investigating seasonal changes in arctic soil microbiomes. Total RNA-seq can be used to address such issues [11–13].

RNA-seq is also well-established for analyzing expressed genes. This "transcriptome analysis" is typically performed with mRNA enriched with complementary DNA (cDNA). rRNA is present in much higher amounts than mRNA [14]. However, current sequencing technology can distinguish mRNA, even in the presence of relatively large amounts of rRNA. Presently, no poly-A tail mRNA containing microbial community can be analyzed by total RNA-seq.

In the RNA-seq process, huge amounts of rRNA information are obtained. This information is used for taxonomic analysis. Arctic environmental microbiologists applied RNA-seq to solve DNA contamination issues [11, 12], and rumen microbiologists used the method to evaluate microbial communities composed of bacteria and ciliates [15, 16]. Therefore, several RNA-seq-based analysis methods are available.

Analysis pipeline work with ribo-tag [12, 15] typically uses reads as tag sequences to annotate and quantify bar codes for molecular classification [5]. Short-length reads are used for this analysis, and annotation resolution is limited, e.g., up to the level of order or family. Identifying up to the level of genus or species requires a low-throughput method such as clone library construction.

Conversely, mapping-based RNA sequences use a different principle to annotate and quantify reads [17–19]. In this case, reads are mapped onto reference sequences, such as ssrRNA database contents. Miss-mapping is still possible because of highly conserved sequences among organisms in the stem region, but finer annotation, e.g., genus level, is still possible.

[17–19]. Little study using mock communities is available to compare total RNA-seq and amplicon-seq approaches. [17, 20].

In the present study, a modified mapping-based all RNA information sequencing (ARI-seq) analysis using a mock microbial community was compared with an amplicon-seq analysis pipeline. We constructed contigs with the obtained reads and mapped these reads onto our own in-house total cDNA database. Simultaneously, we divided the reads into possible ssrRNA origin and others. We then expected that ssrRNA origin and "other RNA" (possibly mRNA and other functional RNA) reads are separately mapped in an in-house cDNA database. This simple process is slightly different from ordinary mapping-based RNA sequences in that reference sequences are constructed from their own reads instead of library contents. This approach is expected to add confidence and accuracy because reference sequences are directly generated from obtained reads.

Our results show that specific conditions of analysis are needed and that our method displays genus-level accuracy for taxonomic assignment. A mock community with ten species was correctly and quantitatively reproduced with assignments superior to amplicon-seq.

## Materials and methods

### Mock microbial community DNA and RNA preparation

We used ten strains of evenly mixed cell material (ATCC MSA-2003, American Type Culture Collection). The material includes well-characterized microbial cells of *Bacillus cereus*, *Bifidobacterium adolescentis*, *Clostridium beijerinckii*, *Deinococcus rediodurans*, *Enterococcus faecalis*, *Escherichia coli*, *Lactobacillus gasseri*, *Rhodobacter sphaeroides*, *Staphylococcus epidermidis*, and *S. mutans*. Freeze-dried material was rehydrated with 1 ml of PBS (−) (137 mM NaCl, 2.7 mM KCl, 10 mM $Na_2HPO_4$, and 1.76 mM $KH_2PO_4$) and stored at −80˚C in 100 µl aliquot.

RNA extraction used RNeasy PowerBiofilm kit (QIAGEN) following the manufacturer's instruction. Two 100 µl aliquots were used as starting material. Obtained RNA solutions were eluted with 50 µl of water and mixed into a single tube (100 µl of RNA solution). Obtained RNA concentration was measured with a Qubit RNA HS kit (ThermoFisher). DNA extraction was performed using a DNeasy PowerSoil kit (QIAGEN) by following the manufacturer's instruction. The DNA solutions obtained were eluted with 50 µl of water and mixed into a single tube (100 µl of DNA solution). The DNA concentration obtained was measured with a Qubit DNA HS kit (ThermoFisher).

### Amplicon-seq analysis

DNA the mock microbial community was used for amplicon-seq analysis with 16S small subunit ribosomal RNA (ssrRNA) gene sequences. We selected two hypervariable target region V4 and V3–V4 for the analysis. Amplicon-seq libraries were constructed using the Illumina "16S Metagenomic Sequencing Library Preparation" protocol with some modifications. Briefly, PCR reaction used PCR enzyme "KOD plus" (TOYOBO) and recommended reaction conditions (1.5 mM $MgSO_4$, 0.2 mM dNTP, 1 unit/50 µl KOD plus, and 0.2 pmoles/µl primers). We used a single-step instead of the original two-step PCR procedure. Primers were designed for the V4 region [10] and V3–V4 region [21].

(Bac515F_D501: 5′– AAT GAT ACG GCG ACC ACC GAG ATC TAC ACT ATA GCC TAC ACT CTT TCC CTA CAC GAC GCT CTT CCG ATC T GT GCC AGC MGC CGC GGT AA –3′, Bac806R_D701: 5′– CAA GCA GAA GAC GGC ATA CGA GAT CGA GTA ATG TGA CTG GAG TTC AGA CGT GTG CTC TTC CGA TCT GGA CTA CHV GGG TWT CTA AT –3′)

(BacV3_V4_F_D502: 5′– `AAT GAT ACG GCG ACC ACC GAG ATC TAC ACA TAG AGG CAC ACT CTT TCC CTA CAC GAC GCT CTT CCG ATC TCC TAC GGG NGG CWG CAG` –3′, BacV3_V4_R_D702: 5′– `CAA GCA GAA GAC GGC ATA CGA GAT TCT CCG AGT GAC TGG AGT TCA GAC GTG TGC TCT TCC GAT CTG ACT ACT ACT HVG GGT ATC TAA TCC` –3′) including TruSeqHT index and linker sequences. V4 target and V3–V4 target reactions were amplified as 98˚C for 2 min, 25 cycles of 98˚C for 15 s, 55˚C for 45 s, 68˚C for 1 min, and 68˚C for 6 min. Products were purified with AMpure magnetic beads following the manufacturer's instructions and then eluted with 50 μl of water.

Obtained PCR products were quantified by quantitative PCR (qPCR) by a KAPA Library Quantification Kit Illumina Platform (KAPA biosystems) following the manufacturer's instructions. A 2 nM pool was constructed based on quantification results. This pool was used for Illumina MiSeq sequencing with 5% PhiX spike-in and obtained 250 bp paired-end reads. Obtained reads were analyzed with the QIIME2 pipeline [22] with DADA2 [23] for quality control and taxonomic assignment with a naïve Bayes classifier for annotation [24]. Each target region was specified by primer sequences to train the naïve classifier with silva132_99.fna of the Silva database, release 132 [4]. Annotation was on taxonomy_7_levels.txt in the same database. Obtained sequences were deposited in DDBJ DRA, accession number DRA009985.

## ARI-seq analysis

Obtained total RNA from the mock microbial community was used to construct a total RNA-seq sequencing library with a SMARTer stranded RNA-seq kit (Clonetech) following the manufacturer's instruction. We used 5.8 ng of RNA as starting material, PCR was repeated for 12 cycles, and final products were eluted by 10 μl of water. The obtained sequencing library was quantified with a KAPA Library Quantification Kit Illumina Platform following the manufacturer's instructions. Again, a 2 nM pool was constructed based on quantification results. The pool was used for Illumina MiSeq sequencing with 5% PhiX spike-in and obtained 250 bp paired-end reads. Obtained sequences were deposited in DDBJ DRA, accession number DRA009985.

Obtained reads were trimmed by trimmomatic-0.39 [25] with option "ILLUMINACLIP: TruSeq_LT_HT.fa:5:30:7 MINLEN:100 HEADCROP:6 LEADING:20 TRAILING:20." PhiX sequences were removed by USEARCH 11.0.667 -filter_phix option [26]. Low complexity filtering was performed with USEARCH 11.0.667 -filter_lowc option [27]. Cleaned reads were used in the assembly process using Trinity v2.8.5 with a minimum_contig_length of 500 [28].

Cleaned reads were sorted into ssrRNA and non-ssrRNA reads using SortMeRNA with paired-in or paired-out options [27], respectively. Reference sequences for sorting with Sort-MeRNA were silva-arc-16s-id95.fasta, silva-bac-16s-id90.fasta, and silva-euk-18s-id95.fasta. Sorted ssrRNA reads were used for mapping against Trinity output (Trinity.fasta). Mapping was performed by bowtie2 v.2.3.5.1-linux-x86_64 with options -1 and -2 used to specify paired mapping mode, while option -U and forward and reverse reads were used to specify non-paired mapping mode. Finally, we used the bowtie2 process in "local mode." Resulting SAM files were transformed with samtools into BAM files and sorted. Sorted BAM files were used to obtain counting information by eXpress v.1.5.1-linux_x86_64 [29]. Count data truncated with a custom script to remove reads with fewer than 10 counts.

Annotation for ssrRNA data–query for extracted sequences from "Trinity.fasta" mapped with ssrRNA reads by SortMeRNA–used the QIIME2 feature classifier command [30] in three modes: (1) classify-sklearn (the same method used for amplicon-seq analysis with naïve Bayes classifier that trained by silva132_99.fna of the Silva release 132 database without region specification) [24], (2) classify-consensus-BLAST [consensus taxonomic assignment by BLAST+ (Bl),

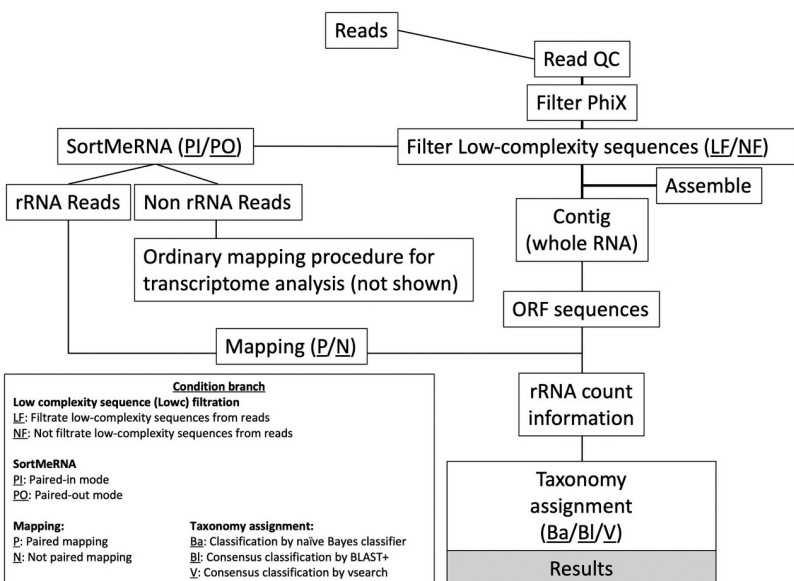

**Fig 1. Analysis scheme.** Flow chart of analysis process of mapping-based RNA-seq analysis to determine microbial community structure. Box indicates branching points in analysis conditions.

first 10 hits] [31], and (3) classify-consensus-vsearch [consensus taxonomic assignment by vsearch (V), top 10 hits] [32].

Count data and annotation information were combined using an in-house script in R statistics software. Finally, we calculated reads per kilobase fragment (rpk) on the basis count data and query sequence length and then calculated relative abundance manually. The analysis scheme is illustrated in Fig 1, and analysis conditions are provided in S1 Table. Log of all scripts and commands will be provided upon request.

## Search conditions for the ARI-seq approach and visualization of results

We used several different conditions for the four steps in the analysis pipeline (S1 Table, explanation of condition branch). First, we used two conditions in the reads qualification step. After trimming and artificial sequence removal, we added a low complexity sequence filtering step. A low complexity sequence is defined as a single nucleotide or short motif repeat in a read that can add noise to the assembly process. However, removal of low complexity sequences mainly affects short reads and can disrupt the assembly of reads to contigs. Therefore, we included the options to perform low complexity sequence filtering (LF) or not (NF).

Second, qualified reads were used for sorting to ssrRNA or non-ssrRNA sequences by Sort-MeRNA, which looks at forward and reverse reads individually; however, result output was paired reads (a set of forward read and reverse read). Hence, two strategies, "paired-in" and "paired-out," in the SortMeRNA program, can be used in the analysis. While a part of paired read was assigned as ssrRNA, the other part was assigned as non-ssrRNA. Both reads (= paired read) were assigned as ssrRNA in "paired-in (PI)" mode and both reads (= paired read) were assigned as non-ssrRNA in "paired-out (PO)" mode. These conditions affect numbers of reads in ssrRNA or non-ssrRNA categories and alter mapping results. Sorted ssrRNA reads were mapped into contigs by bowtie2.

Third, we examined "paired mapping (P)" and "non-paired mapping (N)." Normally, paired reads are used as mate pairs for mapping onto reference sequences (paired mapping).

However, bacterial RNA contains several different gene units as operons. We can expect that half of paired reads can be assigned on reference sequences. Paired reads should be separated into single reads and mapped separately (non-paired mapping). We provided options for these two strategies since they greatly affect mapping.

Reads assembled into contigs are used for taxonomic assignment by a part of the QIIME2 pipeline. Normally, a trained Bayes classifier is used for taxonomic assignment. However, we observed that results are not reliable using this approach. Thus, we included search options with a naïve Bayes classifier using consensus taxonomy classification (Bayes, Ba) with Bl, first 10 hits or consensus classification with V, top 10 hits. "Bl" and "V" are homology search base methods and show better performance than "Bayes (Ba)" conditions, as further discussed in the following section.

Words in parentheses indicate conditions options. Single and combinations of these options represent analysis conditions in the following sections. All possible analysis modes and its abbreviations are shown in S1 Table.

Obtained data were transformed to relative abundance and basic statistical values, such as total relative abundance value of all mock community member and average relative abundance of each mock member. These results are the basis for a cumulative bar plot. Distributions of relative abundance values were visualized with beeswarm plots and box plots. Statistics were calculated with R software. Boxplots show a whisker range of 1.5 × interquartile range and boxes that include first to third quartiles.

## Comparison with other mapping-based RNA-seq analysis

Comparison between our method and the already reported mapping-based RNA-seq analysis was performed with meta-total RNA sequencing (MeTRS) technology [17]. First, we used MeTRS with our mock sequencing data to compare with our method results. Second, we obtained microbiome sequencing data to test MeTRS (SRR5439729 from the SRA database in GenBank) and analyzed it with both our method and MeTRS. MeTRS analysis was performed according to a study [17] with their scripts (https://github.com/normanpavelka/MeTRS) with Silva release 132 ssrRNA database. Some pipeline steps were slightly modified according to issue comments on the GitHub website (https://github.com/normanpavelka/MeTRS/issues/1). Revised codes and the resulting raw data will be provided upon request.

## Results

### Accuracy of taxonomic annotation

Amplicon-seq identified mock community members with high accuracy. Relative abundance (Fig 2) of 99.85% (V3–V4) to 99.90% (V4) for clustered fragments using QIIME2 are assigned correctly to genus. ARI-seq results showed contrasting results among three taxonomic assignment methods. Taxonomic assignment using a naïve Bayes classifier showed low accuracy. Depending on analysis conditions, the relative abundance of mock member genus assignments was only 22.17 ± 16.57%. Especially, "LF–PO mode ssrRNA sequence sorting" and "NF–PI mode ssrRNA sequence sorting" conditions showed very low accuracy (6.96 ± 3.25%, relative abundance of mock member genus). Other approaches correctly showed relative abundance to genus for 37.52% ± 2.70% of community members. Homology search methods (Bl and V) showed relatively high accuracy (94.31% ± 3.49%, relative abundance of mock member genera). MeTRS with our mock community data showed similar accuracy against homology search methods (98.20%, relative abundance of mock member genera).

Taxonomic assignment to "non-mock member" among analysis conditions (S2 Table) indicated that the ARI-seq approach with homology-based taxonomic assignment gave reasonable

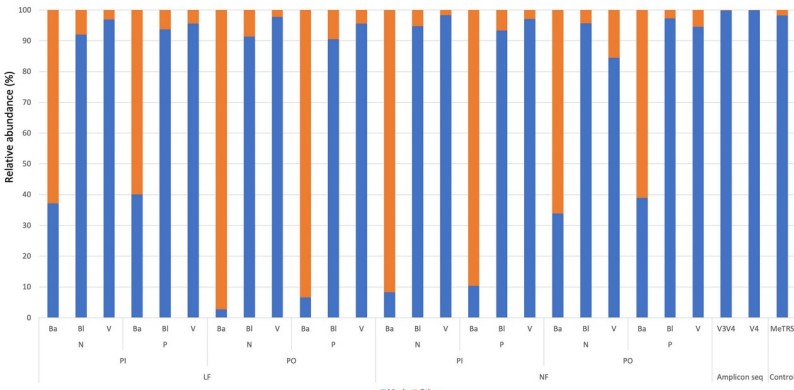

**Fig 2. Accuracy of mock detection among tested methods.** Blue bar indicates a detection rate of mock and orange bar indicates misdetection. Abbreviations of analysis condition in sample names are defined in S1 Table and in the main text.

results, even considering relative abundance chart indications of non-mock member ssrRNA detection. Amplicon-seq detected small amounts of "non-mock members" and detected microbes with no taxonomical relationship with mock members. Homology-based taxonomic assignment of ARI-seq detected few such taxonomically independent sequences, possibly as contaminants (small amounts of *Homo sapience* 18S ssrRNA homolog, and human epidermal bacterium, *Enhydrobacter*, 16S ssrRNA homolog). Furthermore, most detected sequences by homology-based taxonomic assignments for ARI-seq are consensus sequences among kingdom, phylum, order, and family and include mock community members. This finding may reflect conserved regions of ssrRNA sequences that are shared broadly across taxonomically related genera of mock members. For example, genus, *Salmonella*, was detected. This species is closely related to genera, *Escherichia* and *Shigella*. In this context, such results do not indicate miss-assignment. The only exception is detection of plant chloroplast 16S in a few cases; however, detected amounts were low.

Taxonomic assignments by the naïve Bayes classifier for ARI-seq showed many false alignments. The conserved region of ssrRNA may be a problematic identifier. ARI-seq with homology-based taxonomy produced appropriate results compared with amplicon-seq findings.

## Mapping-based total RNA-seq analysis for ssrRNA shows better mock community reconstruction

Relative abundance charts across analysis condition are presented in Fig 3. Except for ARI-seq taxonomy by a naïve Bayes classifier, all analysis conditions accurately detected all ten mock members (also see S3 Table). Amplicon-seq patterns can be uneven, and our results also showed such a pattern. Amplicon-seq with V3–V4 regions showed a significant abundance of *Escherichia–Shigella* and lower abundance of *Bifidobacterium*, *Enterococcus*, *Lactobacillus*, and *Staphylococcus*. Amplicon-seq with V4 region was somewhat less uneven than the V3–V4 amplicon. However, the abundance of *Bifidobacterium*, *Lactobacillus*, and *Staphylococcus* was quite small. In addition, MeTRS showed an uneven pattern, i.e., the pattern was different with amplicon-seq, and it showed a significant abundance of *Bacillus* and a lower abundance of *Bifidobacterium*, *Clostridium*, and *Lactobacillus*. Interestingly, some ARI-seq with homology-based taxonomic assignment showed more likely community structures than amplicon-seq. For example, for "NF–PI mode ssrRNA sequence sorting" with homology-based taxonomic assignment (Bl and V) and for both paired (P) and non-paired (N) mapping modes,

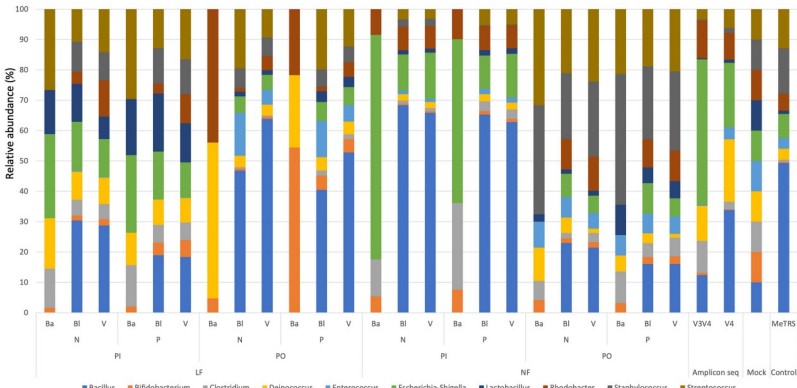

**Fig 3. Accumulative bar chart of relative abundance among detected mock community members.** Color chart is provided in the figure, and abbreviations for analysis methods are defined in S1 Table and in the main text.

abundance pattern is quite even except for a very small abundance of *Enterococcus*. Furthermore, in "NF–PO mode ssrRNA sequence sorting" with homology-based taxonomic assignment (Bl and V) and both paired (P) and non-paired (N) mapping modes, abundance of *Bacillus*, *Staphylococcus*, and *Streptococcus* are relatively large, but a more even pattern is observed than for amplicon-seq and MeTRS results.

Distribution of abundance estimates is provided, as plotted in Fig 4. In "LF–PI mode ssrRNA sequence sorting" mode with homology-based taxonomic assignment (Bl ad V) and both paired (P) and non-paired (N) mapping modes, median abundance of mock members is almost 10%. Amplicon-sequence results showed median abundance of less than 10%, and the distribution of abundance estimates was broader than for ARI-seq. MeTRS showed a similar result with amplicon-seq. The median abundance of MeTRS was similar to that of V4 primer set, and the distribution of abundance was similar to that of V3V4 primer set. Thus, ARI-seq with "LF–PI mode ssrRNA sequence sorting" with homology-based taxonomic assignment (Bl and V) show better reconstruction performance for mock community structure than the amplicon-seq analysis pipeline.

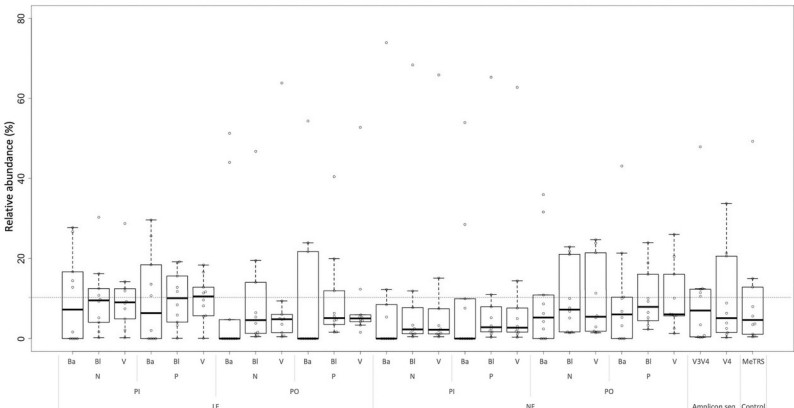

**Fig 4. Scatter and box plots of distribution for relative abundance among mock community members.** The figure shows scatter and box plots. Broken line indicates the 10% line of relative abundance expected from the fraction of each member in the original mock community. Abbreviations of analysis methods are provided in S1 Table and in the main text.

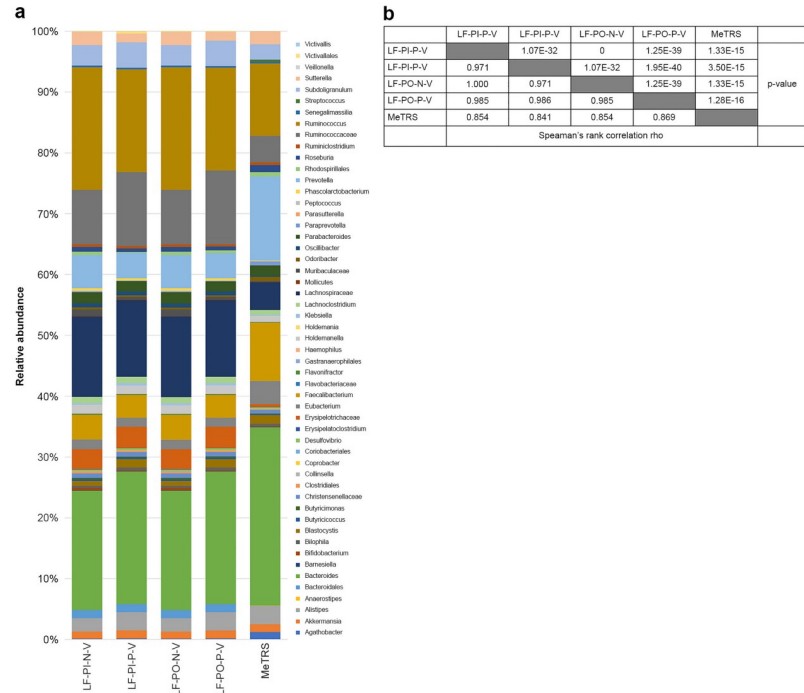

**Fig 5. Genera distribution in our method and MeTRS.** (a) Relative abundance of the 53 genera commonly detected by our method (LF-PI-N-V, LF-PI-P-V, LF-PO-N-V and LF-PO-P-V) and MeTRS in the SRR5439729 data originated from a stool sample. (b) Spearman's rank correlation and *P*-value among the tested methods. Abbreviations of analysis condition in sample names are defined in S1 Table and in the main text.

To test our method with "real-world" data, comparable analysis between our method and MeTRS was performed using published data from a human stool sample. We used SRA data published with MeTRS (SRR5439729) as basal stool microbiome data for this purpose. As shown in Fig 5, the composition of 53 genera commonly detected in this data by our method (LF-PI-N-V, LF-PI-P-V, LF-PO-N-V, and LF-PO-P-V modes) and MeTRS was similar to each other. Spearman's rank correlation and *P*-value indicated that those patterns are significantly similar.

## Discussion

Our ARI-seq approach analysis of microbial populations shows genus-level annotation accuracy and reasonable quantitation among a mix of ten species in a mock community. The traditional total RNA-seq analysis pipeline using the "ribo-tag" concept displays limited taxonomic annotation (class level) [8], and recent work improves annotation only to order or family levels [15, 20]. Our mapping-based method with homology-based annotation showed genus-level accuracy with minor miss-mapping possible in conserved regions (S1 Table).

Results show that our method produces more precise quantitative data than amplicon-seq. Reconstruction of a mock community with ten bacterial species was optimal using (a) "LF–PI mode ssrRNA sequence sorting" with (b) homology-based taxonomic assignment (Bl and V) and (c) both paired (P) and non-paired (N) mapping modes. These features are commonly observed with total RNA-seq methods, and mock analyses using total RNA sequences showed similar results [16, 17, 20]. Indeed, the comparison between our method and MeTRS indicated that some of our analysis conditions showed better results than MeTRS as mock community reconstruction. Furthermore, "real-world" data trial showed that significant similar

community composition was reconstructed from stool RNA-seq data with both our method and MeTRS.

In conclusion, simple mapping-based quantification using ARI-seq displayed better performance for microbiome community reconstruction than amplicon-seq using specific analysis conditions. We optimized our ARI-seq approach by examining four factors in the analysis pipeline–LF, ssrRNA sequence sorting strategy, mapping strategy, and taxonomic assignment methods. Results indicate that removal of low complexity sequences (LF mode), sorting ssrRNA using paired-in mode (PI mode), and using homology-based taxonomic assignment (Bl and V mode) provide optimal reconstruction of a mock community. Total RNA-seq is widely used for meta-transcriptome analysis. The present study indicates that almost the same process can be used for microbiome analysis. Our process should open new opportunities for understanding functional microbiomes with a simple mapping-base analysis pipeline.

## Supporting information

**S1 Table. Branching points for analysis conditions.** Four branching points in the analysis process, with abbreviations of analysis conditions in sample names.
(XLSX)

**S2 Table. False positive detection.** A list of false positive signals and abundance.
(XLSX)

**S3 Table. Quantitative data for mock members.** Reads per kilobase fragment is indicated in RNA-seq data (bundle column of non-paired mapping and paired mapping) and read counts in amplicon-seq data. These data are original data used to calculate relative abundance.
(XLSX)

## Acknowledgments

A Linux-based computational platform used to analyze all data in this work was kindly provided by Yuichi Hongoh (Tokyo Institute of Technology). The author would like to thank Enago (www.enago.jp) for the English language review.

## Author Contributions

**Conceptualization:** Shigeharu Moriya.

**Data curation:** Shigeharu Moriya.

**Formal analysis:** Shigeharu Moriya.

**Investigation:** Shigeharu Moriya.

**Methodology:** Shigeharu Moriya.

**Project administration:** Shigeharu Moriya.

**Software:** Shigeharu Moriya.

**Supervision:** Shigeharu Moriya.

**Validation:** Shigeharu Moriya.

**Visualization:** Shigeharu Moriya.

**Writing – original draft:** Shigeharu Moriya.

**Writing – review & editing:** Shigeharu Moriya.

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
