## [Decision Letter · Decision Letter 0]

29 Apr 2021

PONE-D-20-22602

Simple mapping-based quantification of a mock microbial community using total RNA-seq data

PLOS ONE

Dear Dr. Moriya,

Thank you for submitting your manuscript to PLOS ONE. After careful consideration, we feel that it has merit but does not fully meet PLOS ONE’s publication criteria as it currently stands. Therefore, we invite you to submit a revised version of the manuscript that addresses the points raised during the review process.

We look forward to receiving your revised manuscript.

Kind regards,

Ruslan Kalendar, PhD

Academic Editor

PLOS ONE

Journal Requirements:

2. We note that you are reporting an analysis of a microarray, next-generation sequencing, or deep sequencing data set. PLOS requires that authors comply with field-specific standards for preparation, recording, and deposition of data in repositories appropriate to their field. Please upload these data to a stable, public repository (such as ArrayExpress, Gene Expression Omnibus (GEO), DNA Data Bank of Japan (DDBJ), NCBI GenBank, NCBI Sequence Read Archive, or EMBL Nucleotide Sequence Database (ENA)). In your revised cover letter, please provide the relevant accession numbers that may be used to access these data. For a full list of recommended repositories, see http://journals.plos.org/plosone/s/data-availability#loc-omics or http://journals.plos.org/plosone/s/data-availability#loc-sequencing.

Reviewers' comments:

Reviewer's Responses to Questions

**Comments to the Author**

1. Is the manuscript technically sound, and do the data support the conclusions?

Reviewer #1: Partly

2. Has the statistical analysis been performed appropriately and rigorously? 

Reviewer #1: No

3. Have the authors made all data underlying the findings in their manuscript fully available?

Reviewer #1: Yes

4. Is the manuscript presented in an intelligible fashion and written in standard English?

Reviewer #1: No

5. Review Comments to the Author

Reviewer #1: 

The author reported a mapping-based approach for taxonomic annotation and quantification of RNA sequences. Here are my comments:

1. The method is of interest, but the manuscript in its present form is challenging to understand, making it impossible to assess. Thus, I would ask the author to engage a professional editor to improve the writing of the manuscript.

2. The author compared the performance of his method with Amplicon-Seq. Since multiple mapping-based taxonomic annotation and quantification methods already exist, the author should compare his method with the existing mapping-based methods.

3. The cocktail of 10 bacterial cultures in no way reflects the real-world data. I would use a published dataset along with the dataset generated in the study.

4. BLAST is an acronym, hence replace blast+ with BLAST+.

I suggest improving the writing of the manuscript with a professional editor and compare this new approach with the existing mapping-based approach using a robust real-world published dataset.

6. PLOS authors have the option to publish the peer review history of their article (what does this mean?). If published, this will include your full peer review and any attached files.

Reviewer #1: No

---

## [Author Response · Author response to Decision Letter 0]

26 Jun 2021

Answer for Reviewer’s Comments to the Author

Reviewer #1:

The author reported a mapping-based approach for taxonomic annotation and quantification of RNA sequences. Here are my comments:

1. The method is of interest, but the manuscript in its present form is challenging to understand, making it impossible to assess. Thus, I would ask the author to engage a professional editor to improve the writing of the manuscript.

Author response: Thank you for your comments. I have hired a professional editor to improve the manuscript. In addition, abbreviations are re-organized, and I have added a table in the S1 file for this.

2. The author compared the performance of his method with Amplicon-Seq. Since multiple mapping-based taxonomic annotation and quantification methods already exist, the author should compare his method with the existing mapping-based methods.

Author response: I have added the MeTRS analysis, the only mapping-based method tested by mock and natural sample data. The MeTRS analysis pipeline is introduced, and the data has been analyzed. Results are described in the Results section and in Figs 2, 3, and 4. This method looks slightly better than MeTRS.

3. The cocktail of 10 bacterial cultures in no way reflects the real-world data. I would use a published dataset along with the dataset generated in the study.

Author response: The reported sequencing data was obtained using MeTRS from an original paper that used a human microbiome sample. The sequences were analyzed by the current study’s method and MeTRS, subsequently comparing them as described in the original data. Results are shown in the Results section and in the newly added Fig 5. Statistical analysis showed that both methods are significantly correlated.

4. BLAST is an acronym, hence replace blast+ with BLAST+.

Author response: I have replaced blast+ with BLAST+. Additionally, I have made changes in Figure 1, S1 Table, and the related main text to avoid confusion (abbreviation was changed from “blast” to “Bl”).

I suggest improving the writing of the manuscript with a professional editor and compare this new approach with the existing mapping-based approach using a robust real-world published dataset.

Author response: As mentioned above, I have made the following modifications, with appropriate descriptions.

1. A professional editor has edited the manuscript.

2. The current study’s method was compared with a well-validated RNAseq analysis method, MeTRS.

3. Both mock and “real-world” data were used for comparison between our method and MeTRS.

---

## [Editor Report · Decision Letter 1]

29 Jun 2021

Simple mapping-based quantification of a mock microbial community using total RNA-seq data

PONE-D-20-22602R1

Dear Dr. Moriya,

We’re pleased to inform you that your manuscript has been judged scientifically suitable for publication and will be formally accepted for publication once it meets all outstanding technical requirements.

Kind regards,

Ruslan Kalendar

Academic Editor

PLOS ONE

---

## [Editor Report · Acceptance letter]

7 Jul 2021

PONE-D-20-22602R1 

Simple mapping-based quantification of a mock microbial community using total RNA-seq data 

Dear Dr. Moriya:

I'm pleased to inform you that your manuscript has been deemed suitable for publication in PLOS ONE. Congratulations! Your manuscript is now with our production department. 

Kind regards, 

on behalf of

Prof. Ruslan Kalendar 

Academic Editor

PLOS ONE